# Does Enhanced Structural Maturity of hiPSC-Cardiomyocytes Better for the Detection of Drug-Induced Cardiotoxicity?

**DOI:** 10.3390/biom13040676

**Published:** 2023-04-14

**Authors:** Dieter Van de Sande, Mohammadreza Ghasemi, Taylor Watters, Francis Burton, Ly Pham, Cristina Altrocchi, David J. Gallacher, Huarong Lu, Godfrey Smith

**Affiliations:** 1Global Safety Pharmacology, Nonclinical Safety, Janssen Pharmaceutical NV, B-2340 Beerse, Belgium; 2School of Cardiovascular & Metabolic Health, College of Medical, Veterinary & Life Sciences, University of Glasgow, 126 University Place, Glasgow G12 8TA, UK; 3Clyde Biosciences Limited, BioCity Scotland, Lanarkshire ML1 5UH, Scotland, UK

**Keywords:** cell biology, cardio safety, hiPSC-CM, immature, CiPA, cardiac arrhythmia

## Abstract

Human induced pluripotent stem cell derived cardiomyocytes (hiPSC-CMs) are currently used following the Comprehensive in vitro Proarrhythmic Assay (CiPA) initiative and subsequent recommendations in the International Council for Harmonization (ICH) guidelines S7B and E14 Q&A, to detect drug-induced cardiotoxicity. Monocultures of hiPSC-CMs are immature compared to adult ventricular cardiomyocytes and might lack the native heterogeneous nature. We investigated whether hiPSC-CMs, treated to enhance structural maturity, are superior in detecting drug-induced changes in electrophysiology and contraction. This was achieved by comparing hiPSC-CMs cultured in 2D monolayers on the current standard (fibronectin matrix, FM), to monolayers on a coating known to promote structural maturity (CELLvo™ Matrix Plus, MM). Functional assessment of electrophysiology and contractility was made using a high-throughput screening approach involving the use of both voltage-sensitive fluorescent dyes for electrophysiology and video technology for contractility. Using 11 reference drugs, the response of the monolayer of hiPSC-CMs was comparable in the two experimental settings (FM and MM). The data showed no functionally relevant differences in electrophysiology between hiPSC-CMs in standard FM and MM, while contractility read-outs indicated an altered amplitude of contraction but not changes in time course. RNA profiling for cardiac proteins shows similarity of the RNA expression across the two forms of 2D culture, suggesting that cell-to-matrix adhesion differences may explain account for differences in contraction amplitude. The results support the view that hiPSC-CMs in both 2D monolayer FM and MM that promote structural maturity are equally effective in detecting drug-induced electrophysiological effects in functional safety studies.

## 1. Introduction

In the last decades, preclinical screening for drug-induced cardiac liabilities has proved key to reducing late-stage drug attrition rates [1]. Following the “Comprehensive in Vitro Proarrhythmia Assay (CiPA)” initiative, the focus lies on the effect of compounds on multiple cardiac ion channel types, with the implementation of ion channel screening, in silico modeling, and the testing on human induced pluripotent stem cell-derived cardiomyocytes (hiPSC-CMs) [2,3]. hiPSC-CMs can be obtained by reprogramming adult somatic cells (such as fibroblasts) to a state of pluripotency, following the seminal work by Yamanaka et al. [4]. Through a combination of specific factors, pluripotent stem cells can then be differentiated into cardiomyocytes and used for diverse applications [5]. The hiPSC-CM model is currently widely used in research and the pharmaceutical industry primarily for cardio-safety screening. Medium- to high-throughput techniques are used to screen compounds in development for possible cardiac side effects. Non-invasive methods to measure the function of cardiomyocytes include fluorescent calcium indicators to monitor intracellular calcium, voltage-sensitive dyes or extracellular voltage measurements to follow the transmembrane potential, and movement/force measurements to follow contractility [6,7,8,9,10]. Screening on hiPSC-CMs is currently mostly performed in multi-well plates on which the cells are seeded in a 2D monolayer.

Due to differentiation from a pluripotent state, the hiPSC-CMs have a phenotype closer to that of embryonic than adult cardiomyocytes [11]. This immaturity is observable both on a morphological and an electrophysiological level. Morphologically, the cells are rounder and smaller with a less organized sarcomere structure, while the electrophysiological immaturity consists of a more depolarized and unstable resting membrane potential (RMP), caused in part by a reduced functional expression of the inwardly rectifying potassium channel (Kir2.1) [12,13]. Additionally, relevant expression of HCN channels, responsible for the pacemaker current (*I*_f_) that drives the automaticity in sinoatrial node cells, contributes to the spontaneous firing of action potentials (APs) in hiPSC-CM [14]. Such differences in ion channel expression between adult CMs and hiPSC-CMs [15] represent a challenge to the translation of electrophysiological studies from hiPSC-CMs to the adult heart. However, despite these limitations, commercially available hiPSC-CMs have been shown to be a sensitive phenotypic model for the detection of cardiotoxic effects of compounds [8]. While the model is currently extensively used, it is worthwhile to ask whether attempts to use a more mature phenotype would have additional benefits in detecting effects relevant to the adult heart. Maturation of the hiPSC-CMs phenotype has been achieved using several different methods, including prolonged culture time, sustained electrical stimulation [16,17], different kinds of cell seeding substrates [18,19], inclusion of non-cardiac cells within the culture, and culturing within 3D structures [20,21]. The use of an alternative seeding substrate is further investigated in this study with the use of hiPSC-CMs cultured on commercially available Matrix Plus plates from StemBioSys Inc. (San Antonio, TX, USA), which generate a more mature cell structure [18,19,22] with potentially higher sensitivity to electrophysiological effects [18]. We therefore decided to determine if monolayers seeded on the Matrix Plus model (MM) would show a different response to a drug set designed to assess aspects of cell maturity versus the standard fibronectin coating (FM).

## 2. Materials and Methods

### 2.1. Human-Induced Pluripotent Stem Cell–Derived Cardiomyocyte Cell Culture iCell^2^

Cardiomyocytes (FUJIFILM Cellular Dynamics, Madison, WI, USA) were kept at −190 °C and prepared for culture as per the manufacturer’s instructions. Cell donor (01434) was registered with the ethics committee for research uses (NICHD-NIH, USA, with approval number N-01-HD-4-2865).

Standard fibronectin vs. Matrix Plus well coating: The cells were cultured using two different coatings in a humidified incubator at 37 °C with 5% CO_2_. First, 96-well glass-bottomed plates (MatTek, Ashland, MA) were coated with fibronectin (10 mg/mL in PBS supplemented with Ca^2+^ and Mg^2+^; Sigma, St. Louis, MO, USA) (defining FM). Second, CELLvo^TM^ Matrix Plus (StemBioSys, San Antonio, TX, USA), which were pre-coated on 96 well glass bottom plates and kept at 4 °C. On the day of plating, CELLvo^TM^ Matrix Plus plates were transferred to room temperature, rehydrated with phosphate buffer saline (PBS), incubated for 1 h at 37 °C, and washed 2× with PBS. Cells were introduced at a density of 50,000 cells/well to allow the formation of a confluent monolayer in each well (defined as MM). The maintenance protocols followed the manufacturer’s instructions and used the iCell^2^ Cardiomyocytes Maintenance media for media changes every 2 days. Experiments were performed between days 5 and 6, as recommended by the manufacturers. Before beginning an experiment, cells were washed in serum-free media (SF media) (Fluorobrite DMEM, Gibco, Thermo Fisher Scientific, Horsham, UK).

### 2.2. Drug Treatment

Drug identity and concentration were hidden from the laboratory personnel for the duration of the experiments and subsequent analysis. Eleven reference compounds were used in the study (Appendix A). All of them are well-known compounds, and their effect on hiPSC-CMs has already been extensively investigated.

The drug powders were dissolved in DMSO, and four stock concentrations at 1000× the final target concentration were prepared in DMSO. For the experiment, an intermediate concentration 2× the target solution was prepared in SF media. The addition of drugs to the cells was accomplished by replacing 50% of the well volume with the intermediate 2× solution. The same procedure was done for vehicle control using DMSO. The experimental number *n* = 8/concentration/testing compound in the FM or MM condition, and *n* = 36 for each positive or negative control in the FM or MM condition.

Light protection precautions were taken during stock drug preparation, such as the use of amber vials and silver foil covers. All the drugs (stocks and target concentrations) were prepared in glass containers to minimize drug adsorption to and/or absorption from plastic.

### 2.3. Membrane Potential Signals from hiPSC-CM Using Voltage Sensitive Dyes

Cultures of hiPSC-CMs (5–6 days post-plating) were treated with FluoVolt dye (1:1000, Invitrogen, Cat# F10488) in SF media and incubated for 20 min at 37 °C with 5% CO_2_ and 80% humidity. Multi-well plates were placed in the environmentally controlled stage incubator (37 °C, 5% CO_2_, water-saturated air atmosphere) (Okolab Inc. Burlingame, CA) of the CellOPTIQ^®^ platform (Clyde Biosciences Ltd., Glasgow, Scotland). The FluoVolt signal was recorded from a 0.2 mm × 0.2 mm area using a 40× (numerical aperture: NA 0.6) Excitation wavelength was 470 ± 10 nm using a light-emitting diode (LED), and emitted light was collected by two photomultipliers (PMTs) at 510–560 nm (channel 1) and 590–650 nm (channel 2), respectively. LED, PMT, and associated power supplies and amplifiers were supplied by Cairn Research Ltd. (Kent, UK) and digitized at 10 kHz and stored on a computer hard drive. In the case of FluoVolt-based action potential recordings, the short wavelength (channel 1, 51–560 nm) was analyzed for action potential features. A summary of the CellOPTIQ^®^ configuration is shown in Figure 1.

### 2.4. Voltage Recordings

The dye FluoVolt is loaded into the surface membrane (outer leaflet), and the dye remains within the membrane for many dots, allowing intermittent voltage measurements over the experimental period. In this study, measurements were restricted to within 30 mins of application of the drug. The measurements were made on the same area before/after drug/DMSO addition. As reported in other studies, there were minimal effects of the DMSO vehicle (0.1%) on the function of all electrophysiological and contractility parameters. Baseline spontaneous electrical activity and the associated contractility signal were recorded by capturing 10 s segments of fluorescent signal and video for 2D plates and 15 s for organoids from each well prior to compound addition. Drugs were tested at four concentrations in *n* = 8 independent replicates (i.e., 8 wells from a single plating) at each concentration. A vehicle control was included for each drug. A 10 s or 15 s recording was then taken 30 min after exposure to the drug or vehicle with only one concentration applied/well. Offline analysis was performed using proprietary software (CellOPTIQ^®^). The following (averaged) parameters were obtained from the AP recordings: cycle length (CL, ms); rise time (TRise, ms) between 10–90% of the AP upstroke; and AP durations (APD, ms) from 10–90% repolarization at 10% intervals. Figure 1c shows an example recording of APs (10 s), and Figure 1d shows the result of averaging a train of APs and the main parameter obtained.

### 2.5. Contractility Measurements Using Video Imaging

Contractility was recorded via HCImage Live (Hamamatsu Corporation, Bridgewater, NJ, USA), which controlled a Hamamatsu ORCA-Flash4.0 V3 Digital CMOS camera at 100 fps (1024 × 2048-pixel resolution). Short periods (10 s or 15 s) of video were analyzed using the ContractilityTool application (developed by F.L. Burton), which is an implementation of the MUSCLEMOTION algorithm. This application extracts several parameters from the resultant motion signal, including the average spontaneous frequency and amplitude, the duration at 50% of the contraction transient amplitude (CD_50_), the time to contract (from 10% to 90% of the contraction amplitude), and the time to relax (from 90% to 10% of the contraction amplitude).

### 2.6. Gene Expression

RNA was extracted from cells using the Agilent RNA 6000 Nano Kit (Agilent Technologies, California, USA) as per the manufacturer’s instructions. RNA quantity and quality were assessed using a NanoDrop spectrophotometer and an Agilent 1000 bioanalyzer. RNA sequencing libraries were prepared using the NEBNext Ultra II Directional Library Prep Kit and following the manufacturer’s protocol (NEB, Ipswich, MA, USA). RNA-Seq was performed on the Illumina next-generation sequencing platform.

*Quantification of RNA-seq data:* Transcript expression was quantified using the RNA-seq pipeline in Qiagen OmicSoft Studio V11.7 (Qiagen, Hilden, Germany). Target transcripts were derived from genome assembly Human.B38 from RefGene20210812. Transcript-level quantifications were transformed into gene-level count estimates. The counts were then normalized to a gene-level transcript per million (TPM) value, which was then used as the gene-level abundance estimates used in the analysis.

### 2.7. Data Analysis and Statistics

Kolmogorov-Smirnov tests were used to determine whether the data were normally distributed based on a least-squares fit to a normal function (Origin version 9, OriginLab Corp., Northampton, MA, USA). Statistical analysis was performed using Dunnett’s test following ANOVA to allow the comparison of several treatments with a single control. Statistical significance was designated as ** p <* 0.05. A statistical comparison of electrophysiological and mechanical biomarkers was made using an unpaired *t*-test of the change in the parameter caused by the drug or DMSO.

## 3. Results

### 3.1. Baseline Parameters

Upon seeding of the iCell^2^ cardiomyocytes hiPSC-CM in 2D monolayers, following the company guidelines, a visual distinction could be made between the fibronectin and Matrix Plus coated plates. As previously reported, the fibronectin-coated plates resulted in a monolayer of approximately circular cells arranged randomly. The overall structure of the monolayer was assessed visually and from videos on which contractility measurements were based (see Appendix A). No detailed studies of morphology were undertaken in the present study. The monolayer on the Matrix Plus coated plates showed a more organized monolayer with an elongated morphology and arranged such that the cell’s major axis is oriented in a single direction. Statistically significant, but functionally small differences in baseline parameters were observed between the models, as documented in Figure 2 and Table 1.

The average cycle length of the 2D monolayer on the fibronectin coated plates (1457 ± 7.7 ms) (FM) was significantly different from those on the Matrix Plus plates (1343 ± 6.9 ms) (MM), but this difference is only ~8%. Additionally, the rise time values were within ~8% but again, the difference was statistically significant, as were the values of APD30 and APD90, but the difference between the two groups was only 6–7%. Contractility measurements showed that the two 2D models have similar contraction kinetics, as the contraction duration was not significantly different (to within 2%), but the amplitude of contraction showed the largest difference between the two groups. the MatrixPlus CellVo^®^ group (MM) with an amplitude that is ~50% of that recorded on the fibronectin matrix (FM).

However, cardiac cells in the MM model look more “mature” than those in the FM model in terms of morphology (see Appendix A).

### 3.2. Positive Controls and Vehicle

All drugs were dissolved in DMSO with a final concentration of 0.1%; therefore, a vehicle control was included to determine if 0.1% DMSO influenced the measurements. No significant changes in cycle length, AP rise time (Trise), APD_30_, and APD_90_ were observed, nor were there any significant effects on contractility (contraction amplitude or duration) (Appendix A).

As shown in Figure 3, a set of positive control drugs were chosen based on their common use in multiple applications, including the current 2D screening model, which is routinely used at Janssen Pharmaceutica N.V. Nifedipine (100 nM), a calcium channel antagonist, shortened the APD_90_ and the duration of contraction for the 2D monolayers on fibronectin and Matrix Plus. Dofetilide (10 nM), a hERG blocker, significantly prolonged the APD_90_ in both 2D models by comparable amounts, and isoprenaline (300 nM), a well-known β-adrenergic agonist, increased the spontaneous beat rate and decreased the APD_90_ and associated contraction duration in the 2D systems equally. A further control was used in the blinded study: cetirizine, a histamine-1 receptor antagonist, was used as a negative control across a range of concentrations. The drug normally has no effect on the electrophysiological profile of cardiac cells around the clinical dose. No change in the voltage or contractility parameters were observed with this drug in this study (Appendix A).

### 3.3. Compound Effects across Preparations

The 11 test compounds chosen are routinely used as reference substances within the pharmaceutical industry, and some are also listed in the CiPA compound list. They range from hERG blockers to beat rate modifiers and are well tested in high throughput screening assays at Janssen Pharmaceutica N.V. A sample of the effects of 11 reference drugs are summarized in Table 2.

The effects of the reference drugs on the electrophysiology and contraction are shown below to highlight the type of data acquired. With respect to compounds that inhibit potassium channels, two compounds were selected from the list, and the data are shown in Figure 4. The first compound, quinidine, is a known hERG ion channel blocker and, at higher concentrations, a sodium channel blocker. Figure 4a shows that in both the 2D monolayer on fibronectin matrix (FM) and the monolayer on Matrix Plus (MP), the APD was significantly changed to a similar extent. Cycle length, AP rise time (Trise) and APD_90_ all increased, effects consistent with inhibition of repolarizing currents, specifically *I*_Kr_ and depolarizing currents (*I*_Na_). Higher concentrations resulted in the monolayers becoming electrically quiescent, consistent with inhibition of the sodium current (see Appendix A). The other compound tested was PA-6, which is a selective Kir2.1 channel blocker. This channel is responsible for the stabilization of the resting membrane potential and should be highly expressed in mature cardiomyocytes. PA-6 induced statistically significant changes in the main electrophysiological parameters, such as a mild APD_90_ prolongation, but these effects are not physiologically relevant. Sample records and the concentration dependence of the effects of quinidine and PA-6 are shown in Figure 4.

The effects of levosimendan, a calcium sensitizer, on the two hiPSC-CM models, as an example, are shown in Figure 5 (data in Appendix A). Over the concentrations studied (0.01–1 uM), the drug has no physiologically significant effects on electrical or contractile activity in the two 2D monolayer models. In both formats (FM and MM), contractile amplitude increased at higher concentrations of the drug, but the effects were not reliable enough to be statistically significant. Despite the difference in absolute changes in contractility between the two experimental groups (Table 1), the relative effect of the drugs on inotropy was comparable.

Compounds can also alter the beat rate of cardiomyocytes, and this effect can be induced by targeting different ion channels/processes. Carbachol is a muscarinic acetylcholine receptor agonist that is known to reduce the beat rate of nodal tissue (sino-atrial and atrio-ventricular) and hiPSC-CMs [23]. In the current study, this was evident in both 2D monolayer models as a concentration-dependent increase in cycle length (Table 2 and Appendix A). Oxotremorine, just as carbachol, is a muscarinic agonist and therefore should reduce the spontaneous beat rate. This was observed in both monolayer models to comparable degrees (and there is full data in Appendix A). Isoprenaline is a β1 adrenergic agonist and increases the beat rate of hiPSC-CMs [24] as well as the intrinsic rate of the sino-atrial and atrio-ventricular nodes. The effect on spontaneous rate was clearly observed in the monolayer models from the voltage signal (see Figure 6 and Table 2, and full data in Appendix A). The APD_90_ was significantly decreased in the 2D monolayer groups to equivalent extents.

Diltiazem, a calcium-channel blocker, induced an increase in the beating rate as well as a shortening in APD_30_ and APD_90_, as expected from an inhibition of calcium channels (Figure 7 and Appendix A). Dobutamine is a β1 adrenergic agonist and would be expected to result in an increase in the beat rate of the hiPSC-CMs. When tested on the monolayer models, both showed a comparable concentration-dependent decrease in cycle length (i.e., increased spontaneous rate), while at higher concentrations a number of monolayers became electrically quiescent (Figure 7 and Appendix A).

A statistical comparison of electrophysiological and mechanical biomarkers was made using an unpaired *t*-test of the change in the parameter caused by the drug or DMSO. The data for a subset of tests are shown in Appendix A. The majority of the parameters showed no significant differences between the two matrix types (Appendix A). There were some biomarkers that did show statistical significance, but the extent of the change observed was generally very small and without functional significance. One result with potential physiological significance is the increase in APD associated with carbachol in the CellVo matrix (MM), which is generally 2× that of the conventional matrix (FM) (see Appendix A).

### 3.4. Gene Expression

Earlier studies suggested that the novel Matrix Plus Cellvo™ substrate was able to mature the structural phenotype of the iPSC-CM compared to the conventional fibronectin matrix [18]. We therefore aimed to understand if any change in the gene expression would occur when a different culture substrate would be used. As the two models (FM and MM) used the same cardiomyocytes (iCell^2^, Fujifilm CDI), samples of cells from the two models were used to examine gene expression from a select list of genes known to directly code for key cardiac-specific proteins.

As shown in Figure 8, using a principal component analysis, the two models are separated by component 1 on the *x*-axis. Key-genes of cardiac ion channels, cardiac pumps, receptors, and proteins regulating cardiac functions are expressed in both models (Table 3). The RNA copy numbers of cells cultured on fibronectin matrix (FM) and Matrix Plus (MM) differ minimally from each other. Higher expression of the Kir2.1 channel, responsible for the inward rectifying potassium current (*I*_K1_) controlling the negative resting potential of the membrane potential of mature ventricular cells, would reflect a more mature phenotype. However, there was no significant difference in the RNA expression of this channel (fibronectin matrix (FM): 43 ± 2.4 TPM; Matrix Plus Cellvo™ matrix: 31.2 ± 2 TPM—Table 3). The relatively robust and comparable expression of Kir2.1 was present in parallel with the significant expression of the pacemaker ion channel (HCN4). In mature ventricular myocytes, this ion channel expression is very low, which explains the stable diastolic membrane potential. Thus, the use of Matrix Plus did not significantly affect the relative balance of Kir2.1 and HCN4 in favor of a more mature cardiomyocyte profile.

## 4. Discussion

In this study, the electrophysiological and contractile characteristics of two groups of hiPSC cardiomyocytes were studied at baseline and in response to a series of 11 drugs with well-established actions. The novel cellular matrix (Maturity Matrix Cellvo™) has been reported to enhance the differentiation of myocytes beyond that achieved on a standard matrix [18]. This study directly compared the electrophysiological and contractile responses from cultures derived from a common sample of commercial iPSC-CMs plated either on the novel matrix or on the standard fibronectin coating. The data shows that although some of the baseline electrophysiological parameters were significantly different between the two experimental groups, the differences were small (~5%) and were not thought to reflect major functional differences of the hiPSC-cardiomyocytes (Table 1). The time course of contraction was also similar between the two groups, but the amplitude of contraction was significantly reduced (~50%). The RNA expression profile of 22 proteins indicates that the relative gene expression of cardiac ion channel proteins and others associated with E-C coupling were not significantly affected by culturing on the specialized matrix (Table 3).

Inhibition of hERG (I_Kr_) blocker by dofetilide caused statistically identical effects on repolarization between the two experimental groups. A similar effect on repolarization was observed on addition of quinidine, accompanied by an increase in AP rise time, an affect attributed to the known Na-channel blocking actions of the drug [25]. The similar expression levels of genes KCNH2 and SCN5A in the two experimental groups are consistent with the comparable electrophysiological data.

Interesting to note is the lack of effect of the selective blocker of the Kir2.1 channel (PA-6) in both experimental groups; this channel is responsible for the background inwardly rectifying potassium current (*I*_K1_). Yet both monolayer models had similar RNA expression signals for Kir2.1 and Kir 2.3 channels, suggesting that despite the RNA signal, there were minimal levels of functional channels in the cells from both experimental groups. The response to L-type Ca^2+^ channel inhibition in two monolayer models was similar for both nifedipine and diltiazem (Table 2). Again, this is consistent with the similar RNA expression levels of CCACNA1 in the cells from the two experimental groups (Table 3).

Mimetics of the parasympathetic and sympathetic nervous systems were tested in an effort to detect functional maturity as a consequence of the use of the specialized matrix. The addition of acetylcholine or oxotremorine causes a decrease in the spontaneous rate and very small changes in the APD (<10%) in both groups. The simulation of muscarinic (M2) receptors in the absence of β-adrenergic agonism would not be expected to cause significant changes to ventricular muscle electrophysiology; the minor changes observed may reflect a small population of atrial-like myocytes within the human-iPSC [26]. Note that the larger increase in APD in response to carbachol observed in the CellVo matrix compared to fibronectin suggests a stronger atrial phenotype of the cells on this matrix. This result will require further testing to establish the reproducibility and extent of these effects.

Again, the similarity of the sensitivity to these muscarinic receptor agonists between the two experimental groups is consistent with the similar expression levels of the associated RNA (CHRM2, Table 3). The β-adrenergic agonists dobutamine and isoprenaline caused distinct electrophysiological effects that were paralleled in the two experimental groups. There was an increase in the spontaneous rate (decreased cycle length) in both cell systems, indicating an increased activity of the funny channel (*I*_f_) by cAMP, as reported in nodal cardiac tissue and previously reported in iPSC-CM systems [24]. The significant and comparable expression of HCN4 in both FM and MM groups indicates that the degree of electrophysiological differentiation was similar in both groups. Maturation would be expected to alter the relative ratio of expression of HCN4 and Kir2.1 towards a more dominant expression of the latter, but no evidence of this shift in cells on the MM matrix was observed. The second electrophysiological effect was the pronounced concentration-dependent decrease in APD. This is a well-established response to adrenergic stimulation in ventricle electrophysiology and is due to the activation of the slowly inactivating inwardly rectifying potassium channel current (*I*_Ks_) carried by the combined channel composed of KCNQ1 and KCNE1 subunits [27]. This effect is mediated by A-kinase linked phosphorylation. The statistically identical APD responses in both experimental groups are consistent with the matched relative expression profiles of these channel subunits and matched signaling systems. Another feature of interest is the contractile responses to adrenergic stimulation, which in the adult ventricle are characterized by an increase in contraction amplitude (positive inotropy) and a shortened time course of contraction (positive lusotropy). In both experimental groups, decreased contraction duration suggests that the lustotropic actions of adrenergic stimulation were equally present, but in neither group was there a significant positive inotropic response (Figure 6). This form of adrenergic contractile response has been previously reported in hiPSC-CMs (e.g., [24]) and is thought to be based on the poorly developed sarcoplasmic reticulum systems within these cells, a feature that is unaltered by culturing on the specialized matrix system (MatrixPlus Cellvo™).

The contractile characteristics of the two experimental groups indicated that the amplitude of the signal associated with the twitch measured from cells on the MatrixPlus Cellvo™ was ~50% of that measured from cells cultured on fibronectin matrix. The reason for this is unknow; RNA profiles of proteins associated with contractive proteins (MYL2, MYL4, MYL6, MYL7, and TnnC1) were not significantly altered. Previous studies indicate that the contraction of monolayers of hiPSC-CMs depends on the flexibility of the underlying matrix and the balance of cell-to-cell and cell-to-matrix adhesions [24]. These factors may be quite different on the MatrixPlus CellVo™ system compared to the fibrinogen matrix [18] and may account for the differences in the amplitude of the signal reported here. Further work is required to investigate the difference in the contractile signals observed.

The positive inotropes levosimenadan and digoxin were both ineffective in increasing the amplitude of contraction, with both experimental groups producing statistically identical results. Levosimendan is known to mediate effects via direct sensitizing actions on cardiac myofilaments [28] and small (<20%) change were observed at higher concentrations in both groups but this failed to reach significance. Digoxin decreased APD in both groups consistent with raised intracellular Na^+^ levels and altered activity of the electrogenic Na/Ca exchanger but failed to increase inotropy and at higher concentrations both groups displayed electrical quiescence while remaining electrically active (Appendix A). The reason for this range of responses is not known, but it is consistent with a poorly developed sarcoplasmic reticulum that becomes spontaneously active at high intracellular Ca^2+^ concentrations and generates contraction waves in the absence of action potentials [29].

## 5. Conclusions

The results in this study indicate that the use of a specialized culture system designed to increase the apparent maturity of hiPSC-CMs in culture (MatrixPlus CellVo™) generated cardiomyocytes with different contractile characteristics but very similar electrophysiological features. There were only minor differences in baseline electrophysiological biomarkers and no difference in response to the 11 standard reference drugs with known pharmacological effects on the heart. Features linked to maturity (spontaneous rate and inotropic responses) were not significantly different, and both systems were equally sensitive and valid as a testing platform. Further investigation is required to uncover features that are a consequence of the more mature morphology previously reported [18].

## Figures and Tables

**Figure 1 biomolecules-13-00676-f001:**
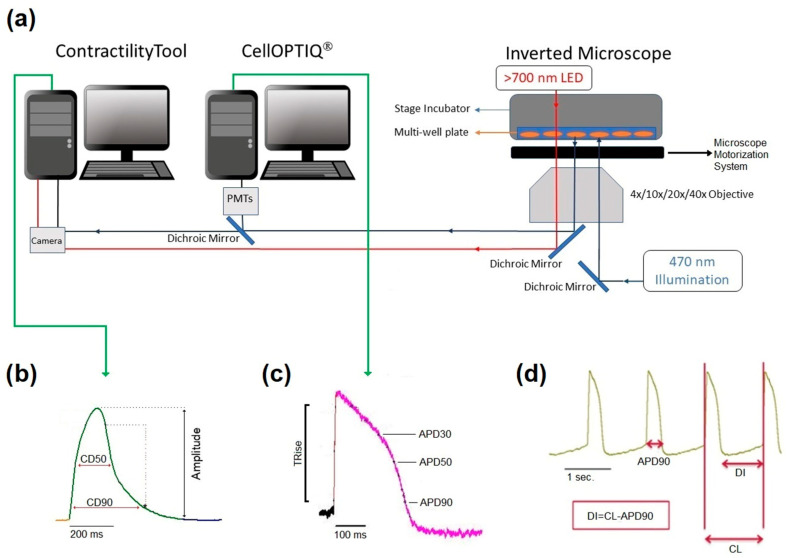
Diagrammatic summary of the CellOPTIQ^®^/ContractilityTool^®^ platform (**a**). Two real-time data acquisition software systems run on separate computers. Contraction of 2D monolayer cultures of hiPSC-CMs is assessed by video recording (100 Hz 1024 × 1024) of a wide field image collected from an Olympus inverted microscope and analyzed using the MuscleMotion algorithm in real-time. Two channels of photometry-based fluorescence signals are acquired by the CellOPTIQ^®^ platform (10 KHz/channel), while illumination and motorized stage position are controlled by digital outputs regulated by the CellOPTIQ^®^ software. Cells are maintained at 37 °C (5% CO_2_) in multiwell plates using a stage incubator. Panel (**b**), example records of monolayer contraction, and corresponding action potential waveform (**c**) from hiPSC-CMs loaded with FluoVolt^®^ voltage-sensitive dye. (**d**) A representative train of APs and the main parameter obtained and recorded with FluoVolt^®^.

**Figure 2 biomolecules-13-00676-f002:**
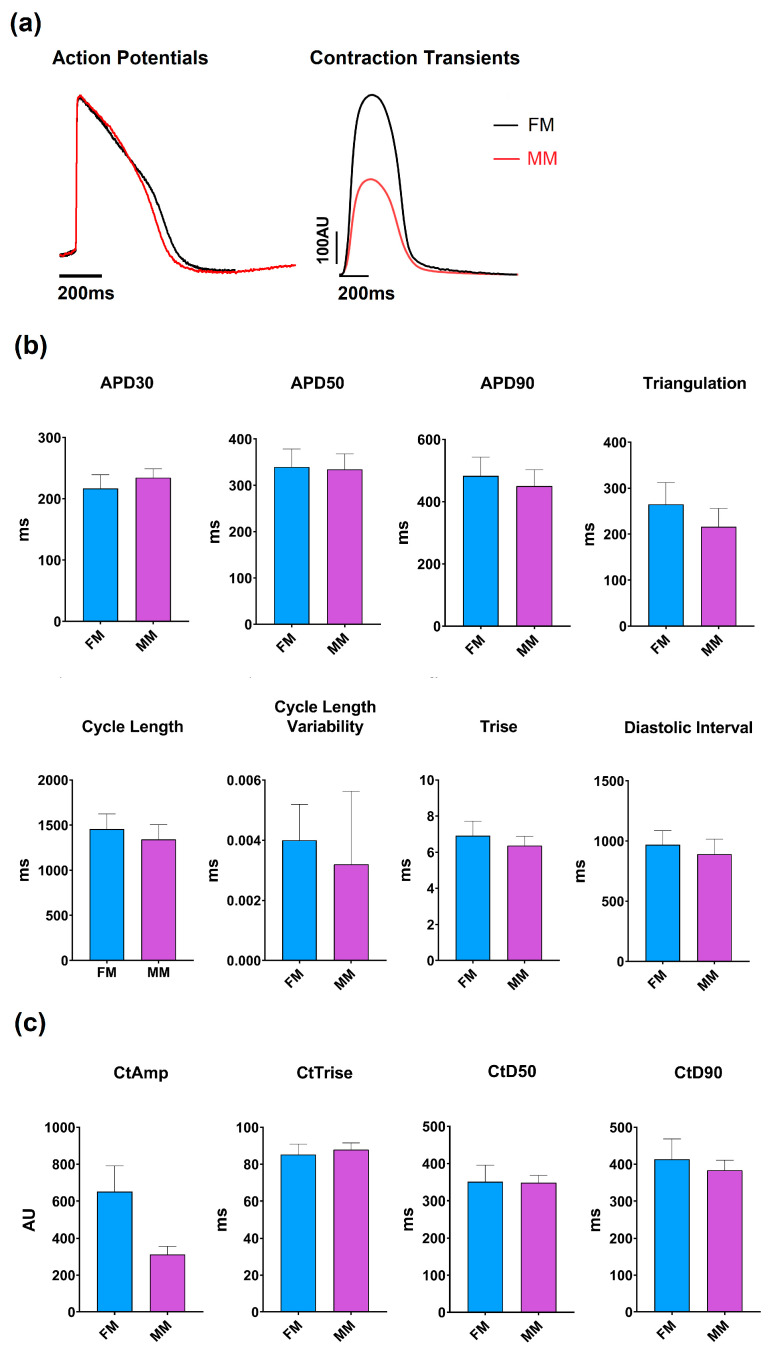
Overview of the models’ base parameters. (**a**) Example traces of the two models are shown for the voltage dye (**left**) and contractility (**right**) recordings: monolayer model with fibronectin (FM) (blue) and Matrix Plus coated plates (MM, violet). (**b**) Mean ± SEM of the key electrophysiological biomarkers; (**c**) Mean ± SEM of the key contractility parameters.

**Figure 3 biomolecules-13-00676-f003:**
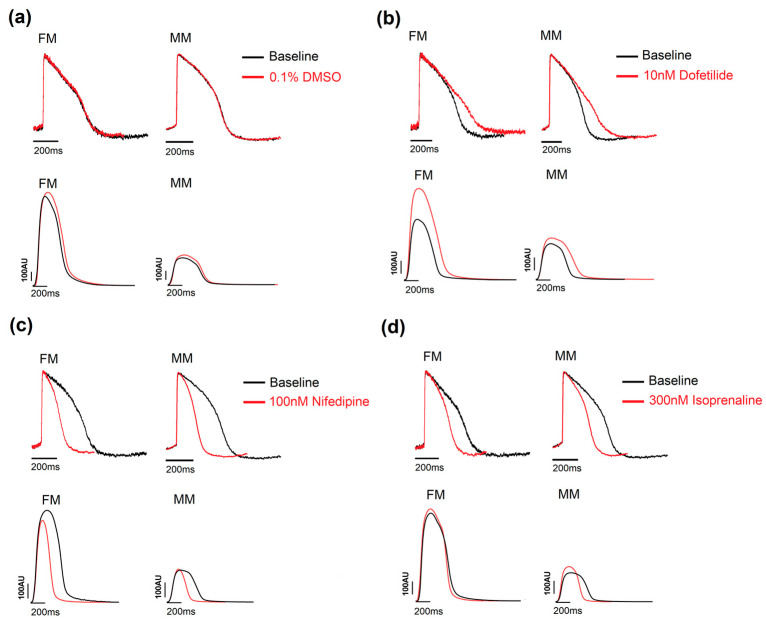
Example traces showing the effect of the three positive controls. (**a**) DMSO 0.1% was used as a vehicle control and had no effect on the traces in the voltage and contractility experiments. (**b**) Dofetilide, a hERG blocker, prolonged the APD in all three models in the two assays. (**c**) Nifedipine reduced the APD_90_ and contraction duration for the 2D models. (**d**) Isoprenaline, a β-adrenergic receptor agonist, increased the beat rate and shortened the duration of the traces in both the 2D models. Monolayer in fibronectin (FM) or in Matrix Plus coated plates (MM).

**Figure 4 biomolecules-13-00676-f004:**
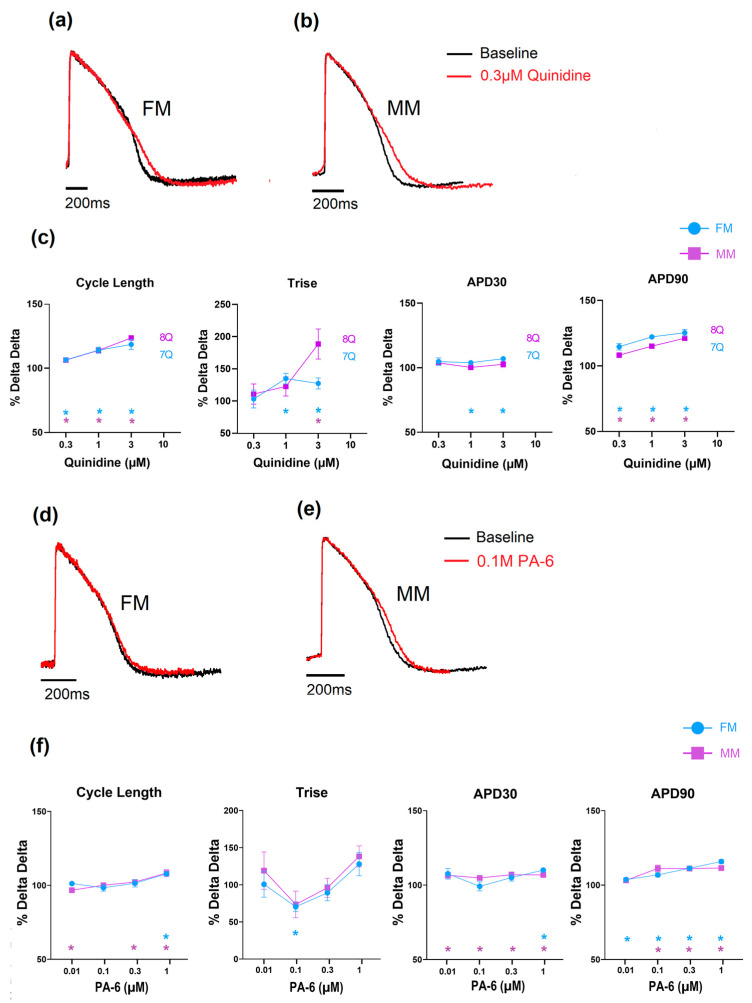
The effect of quinidine and PA-6 on electrophysiology and contractility in the monolayer seeded on fibronectin (FM) and Matrix Plus (MM); (**a**–**c**): effects of quinidine; (**d**–**f**): effects of PA-6. (* *p* < 0.05).

**Figure 5 biomolecules-13-00676-f005:**
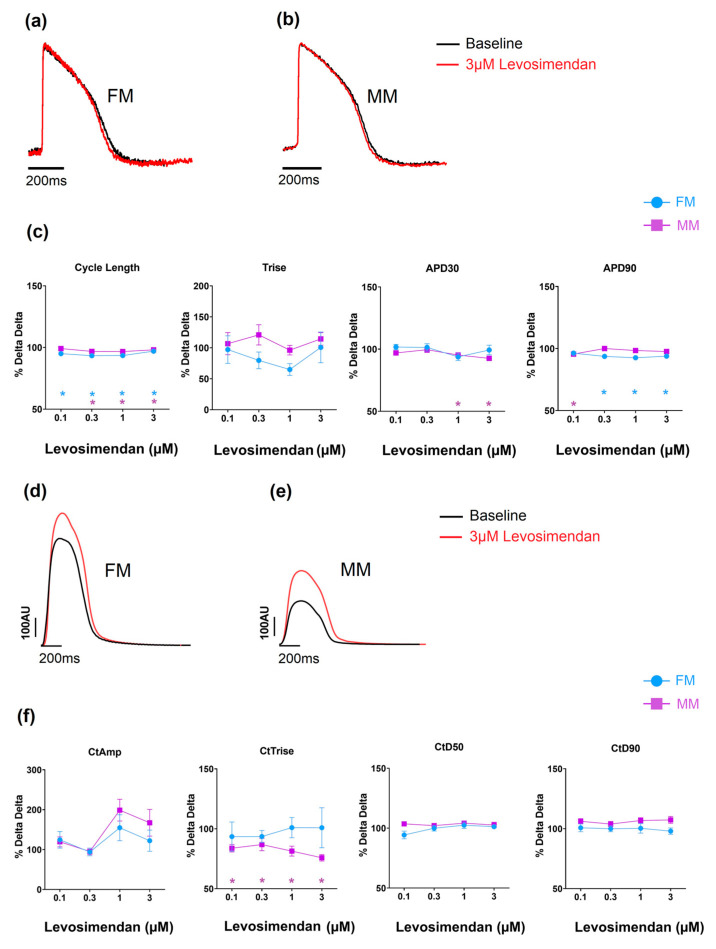
The effect of levosimendan on electrophysiology (**a**–**c**) and contractility (**d**–**f**). * *p* < 0.05 with fibronectin (FM) (blue) and Matrix Plus coated plates (MM, violet).

**Figure 6 biomolecules-13-00676-f006:**
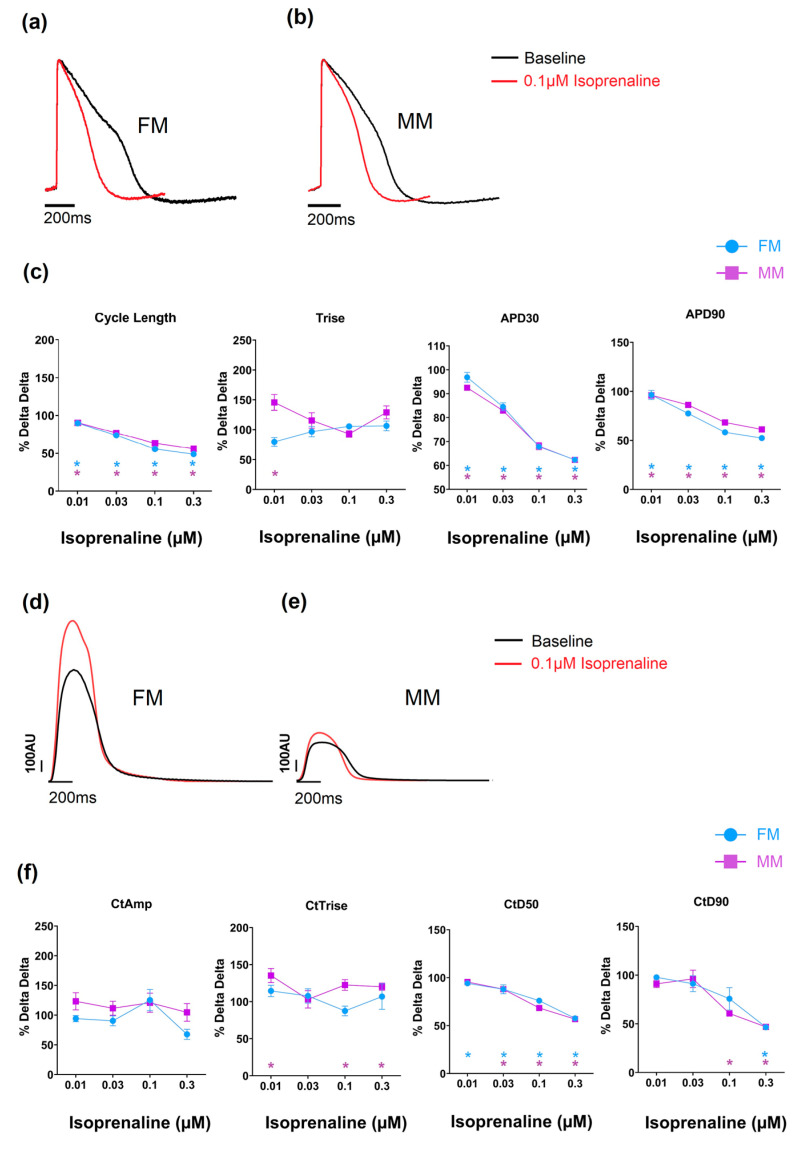
The effect of isoprenaline on the two models on electrophysiology (**a**–**c**) and contractility (**d**–**f**) (* *p* < 0.05). Fibronectin (FM) (blue) and Matrix Plus coated plates (MM, violet).

**Figure 7 biomolecules-13-00676-f007:**
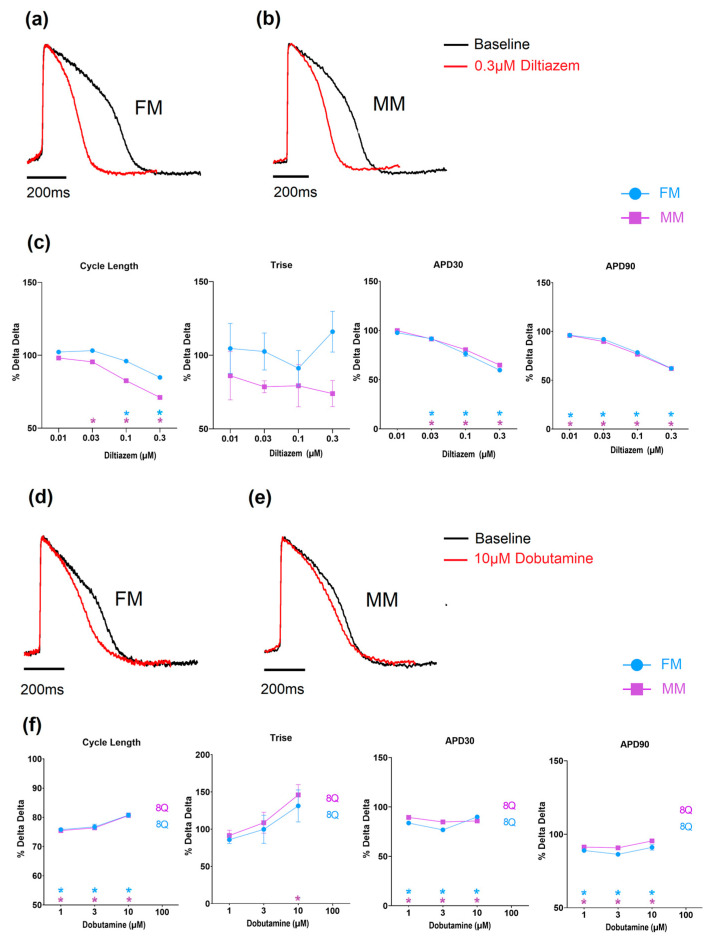
The effect of Diltiazem and Dobutamine on the two models of electrophysiology (Diltiazem: (**a**–**c**), Dobutamine: (**d**–**f**) (* *p* < 0.05)). Fibronectin (FM) (blue) and Matrix Plus coated plates (MM, violet).

**Figure 8 biomolecules-13-00676-f008:**
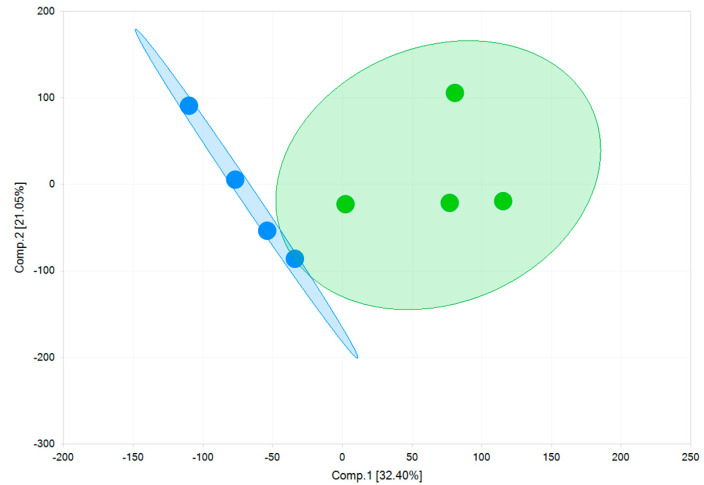
Gene expression principal component analysis of the two models. The first principal component (Comp.1) clearly shows a high degree of variability between two 2D monolayers of fibronectin (FM) (blue) and a 2D monolayer of Matrix Plus (MM) (green).

**Table 1 biomolecules-13-00676-t001:** Overview of the relevant baseline parameters of the two different models. This table represents the most relevant parameters obtained from the two types of biomarkers (electrophysiology and contractility). Cycle length (Cl), rise time (Trise), action potential duration (APD) at 30% and 50%, contractility amplitude (CtAMP) and contractility duration at 50% amplitude (CtD50).

Culture	CL (ms)	Trise (ms)	APD30 (ms)	APD90 (ms)	CtAmp (AU)	CtD50 (ms)
FM	1457 ± 7.7	7 ± 0.1	217 ± 1.4	483 ± 2.6	650 ± 2.9	351 ± 2.3
MM	1343 ± 6.9 *	6 ± 0.1 *	234 ± 1.1 *	450 ± 2.4 *	312 ± 4.7 *	348 ± 1.5

The data are expressed as mean ± SEM (* *p* < 0.01) *n* = 568 wells (FM) and *n* = 574 wells (MM). All parameters apart from contraction duration were significantly different between experimental groups. The monolayer on fibronectin matrix: FM and the monolayer on Matrix Plus: MM.

**Table 2 biomolecules-13-00676-t002:** Overview of the effect of the different compounds at specific concentrations on the relevant parameters of the two models. The data are expressed as mean ± SEM. FM monolayer represents the fibronectin matrix, and the MM monolayer represents the MatrixPlus. * *p* < 0.05 vs. 0.1% DMSO control.

Compound	Culture	CL (%DD)	Trise (%DD)	APD_30_ (%DD)	APD_90_ (%DD)	CtAmp (%DD)	CtD_50_ (%DD)
Quinidine (1 µM)	FM	114 ± 2.4 *	135 ± 8.3 *	104 ± 1.4 *	122 ± 1.4 *	162 ± 50.3	106 ± 1.8 *
	MM	114 ± 1.5 *	123 ± 15	100 ± 1.2	115 ± 0.8 *	99 ± 13.1	97 ± 2.5
Diltiazem (0.1 µM)	FM	96 ± 0.8 *	91 ± 12	76 ± 2.7 *	78 ± 2.2 *	108 ± 33.5	82 ± 1.6 *
	MM	83 ± 0.6 *	79 ± 14.3	80 ± 1.6 *	77 ± 1.2 *	93 ± 6.2	81 ± 1.2 *
Milrinone (10 µM)	FM	103 ± 0.6 *	106 ± 26.8	92 ± 2 *	95 ± 1.1 *	81 ± 5.7	101 ± 1.7
	MM	103 ± 0.2 *	102 ± 7	95 ± 1 *	100 ± 0.9	129 ± 28.5	100 ± 1.4
Levosimenadan (3 µM)	FM	97 ± 1.2 *	101 ± 24.9	99 ± 3.9	94 ± 1 *	122 ± 26.6	101 ± 1.8
	MM	98 ± 0.5 *	114 ± 10.2	93 ± 1.9 *	98 ± 0.9	167 ± 33.5	103 ± 2.3
Digoxin (0.1 µM)	FM	98 ± 0.7 *	100 ± 21.3	84 ± 1.8 *	95 ± 0.6 *	88 ± 6.1	90 ± 1.3 *
	MM	102 ± 0.4 *	127 ± 10.8	91 ± 1.2 *	101 ± 1.3	127 ± 26.2	97 ± 2
Carbachol (1 µM)	FM	114 ± 0.5 *	95 ± 17.7	103 ± 1.6	110 ± 1.1 *	109 ± 13.8	107 ± 3.7
	MM	131 ± 0.7 *	143 ± 17.8	120 ± 2.3 *	118 ± 1 *	114 ± 18.4	118 ± 1.5 *
Cetirizine (3 µM)	FM	98 ± 0.9	122 ± 8.8	99 ± 3.8	92 ± 1.9	112 ± 20.4	98 ± 2.3
	MM	106 ± 2.9	100 ± 8.9	91 ± 1.9 *	93 ± 0.6 *	87 ± 5.3	97 ± 2.4
Oxotremorine (3 µM)	FM	111 ± 3.6 *	132 ± 11.2	107 ± 3.4 *	101 ± 2.8 *	166 ± 17.6 *	110 ± 1.6 *
	MM	123 ± 1.1 *	158 ± 32	115 ± 3.8 *	113 ± 1.2 *	103 ± 7.4	120 ± 2.3 *
Dobutamine (10 µM)	FM	81 ± 0.8 *	131 ± 21.5	90 ± 2 *	91 ± 1.9 *	82 ± 3.2 *	82 ± 2.2 *
	MM	81 ± 0.6 *	146 ± 13.9 *	86 ± 1.9 *	96 ± 0.9 *	98 ± 7.5	77 ± 1.3 *
PA-6 (1 µM)	FM	108 ± 1.4 *	128 ± 15.8	110 ± 1.3 *	116 ± 1.3 *	111 ± 8.1	112 ± 2.5 *
	MM	108 ± 2.4 *	138 ± 14.2	107 ± 1.5 *	111 ± 1.7 *	114 ± 9.8	107 ± 1.2 *
Isoprenaline (0.1 µM)	FM	56 ± 0.6 *	106 ± 4.5	68 ± 0.8 *	60 ± 1 *	125 ± 17.9	76 ± 2.8 *
	MM	63 ± 0.8 *	93 ± 5.5	68 ± 1.5 *	68 ± 1.2 *	121 ± 16.1	68 ± 2.7 *

**Table 3 biomolecules-13-00676-t003:** Gene expression of key cardiac ion channels and receptors in cells cultured on fibronectin (FM) and Cellvo™ Maturity Matrix (MM). The data are expressed as mean ± SEM transcripts per million (TPM).

Gene	Protein	FM (TPM)	MM (TPM)
*ADRB1*	β1 adrenergic receptor	28 ± 10.7	11 ± 2.4
*ATP1A1*	Na^+^/K^+^ ATPase	13,449 ± 1132.2	16,443 ± 1493.5
*CACNA1C*	Voltage-gated Ca^2+-^channel—Ca_v_1.2	4344 ± 265.8	6229 ± 374.5
*CHRM1*	Muscarinic acetylcholine receptor M1	1 ± 0.3	0 ± 0.2
*CHRM2*	Muscarinic acetylcholine receptor M2	1167 ± 133.4	2757 ± 430.8
*HCN2*	hyperpolarization-activated cyclic nucleotide-gated channel 2	379 ± 19.6	302 ± 15.2
*HCN4*	hyperpolarization-activated cyclic nucleotide-gated channel 4	1721 ± 237.7	2352 ± 463.7
*KCNA5*	Voltage-gated K^+^-channel—K_v_1.5	16 ± 8.3	9 ± 4.1
*KCNE1*	K_v_-channels β subunit—minK	81 ± 5.6	84 ± 8.4
*KCNH2*	Human Ether-à-go-go-Related Gene—hERG	2811 ± 133.4	2411 ± 118.3
*KCNJ11*	ATP-sensitive K^+^ channel—Kir6.2	113 ± 7.7	148 ± 24.2
*KCNJ2*	Inward-rectifier K^+^ channel—Kir2.1	43 ± 2.4	35 ± 0.6
*KCNJ4*	Inward-rectifier K^+^-channel—Kir2.3	147 ± 11	149 ± 17.3
*KCNQ1*	Voltage-gated K^+^-channel—K_v_7.1	1006 ± 71.6	925 ± 47
*MYH6*	Myosin heavy chain 6	20,943 ± 1256.1	37,919 ± 1177.9
*MYH7*	Myosin heavy chain 7	218,434 ± 11,851.5	212,866 ± 9265.9
*MYL2*	Myosin light chain 2	67,277.5 ± 2061.8	43,888 ± 1100.6
*MYL4*	Myosin light chain 4	55,748 ± 3432.8	45,843 ± 1119.1
*PDE3A*	Phosphodiesterase 3A	62 ± 6.8	51.5 ± 42.9
*SCN1B*	Voltage-gated Na^+^-channel β1 subunit	47 ± 4.3	31.5 ± 2
*SCN5A*	Voltage-gated Na^+^-channel—Na_V_1.5	8197 ± 406.9	8277.5 ± 618.3
*TNNC1*	Troponin I	60,057.5 ± 2139.7	45,849.5 ± 1828.3

## Data Availability

The raw data supporting the conclusions of this manuscript will be available by the authors upon request without undue reservation.

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
