# Peer review of "Does Enhanced Structural Maturity of hiPSC-Cardiomyocytes Better for the Detection of Drug-Induced Cardiotoxicity?"

_biomolecules, 2023, doi:10.3390/biom13040676_

Round 1

Reviewer 1 Report

The authors addressed a very important problem, namely the effectiveness of using hiPSC-CMs in culture for cardiotoxicity studies depending on the degree of maturation of cardiomyocytes. At the same time, extensive work was carried out comparing the electrophysiological effect of 11 different substances on hiPSC-CMs cultured on conventional substrates coated with fibronectin and on substrates coated with Matrix Plus. The latter, as previously stated by Block, T., et al. should allow faster maturation of hiPSC-CMs. However, the authors found that the electrophysiological characteristics of cells on these substrates differ insignificantly, and the only significant difference is the change in the amplitude of spontaneous contractions, while this amplitude is twice as high on fibronectin.

In my opinion, the work is extremely important, as it shows that the use of Matrix Plus coated substrates is not at all better than conventional fibronectin in terms of electrophysiological studies. However, the title of the article is misleading: the authors showed no differences in the characteristics of the ion channels of hiPSC-CMs cultured on different substrates, without making even a superficial comparison of the maturity of cardiomyocytes. At the same time, it is quite possible that the statement about a significant increase in the degree of maturation of cells on Matrix Plus is an exaggeration or erroneous. I would recommend changing the title of the article.

Author Response

We thank for editor’s letter and reviewers’ comments concerning our manuscript. Those comments are all valuable and certainly improve the quality of manuscript.

According to the right comment of the reviewer 1, we now updated the title as below

“Does enhanced structural maturity of hiPSC-CMs improve drug-induced cardiotoxicity detection?"

Reviewer 2 Report

In the manuscript entitled “Are more mature hiPSC-cardiomyocytes better for the detection of drug-induced cardiotoxicity?”, the authors compared CELLvo to fibronectin if hiPSC-CMs differentially response to drugs. It is interesting to see that electrophysiological properties and contractility can be different, and contractility is weaker on CELLvo matrix that is expected to induce more maturation of hiPSC-CMs. As the positive inotropes were ineffective, the culture condition proposed in the manuscript does not produce “mature” hiPSC-CMs. Thus, what further investigation needed is not the consequence of the more mature morphology but finding better condition(s) in which hiPSC-CMs respond to the positive inotropes. 

1.      Amplitude (y-axis) should not be adjusted 100% for each, instead shown in same AU scale (e.g. Figure 2a, but should be applied to all trace panels). Changes in amplitude are also important information and omitting such information by adjusting Y-axis scale should be avoided. 

2.      Overall concept is whether MM is superior to FM in physiological properties responding to drugs; in that regard, statistical significance should be examined between them, rather than comparing to basal condition of each. I would like to have all the representative traces of both AP and contractility, even they don’t have significant changes. 

3.      There are no N reported. How many biological/technical replicates were used in each study? Another question is reproducibility of the experiments. In this study, the authors recorded 10-15s, applied a drug for 30 min, then recoded again. Can the authors find and record the same cells/area? If so, can the authors obtain reproducible data, especially with FluoVolt that may be bleached after imaging? 

4.      Although authors declared no COI, they use CellOPTIQ platform from Clyde Biosciences, with which 4 of them are affiliated. This is obvious COI. Regarding the affiliations, “Janssen” is missing from #1. 

Minor: 

1.      Line 95-96, (defing as FM) should be (defining as MM). 

2.      In Figure 1, Up90 was define, but not CtTrise that is used in other figures. Are these same? 

3.      While the authors opted not to include morphological features of hiPSC-CMs on CELLvo or fibronectin, it would be helpful to add pictures/immunostaining in the manuscript, even though morphological properties have been reported. It is difficult to judge the morphology in the supplemental video. 

4.      RNA-seq deta is better to be deposited so that other researcher can explore the dataset. 

Author Response

Responses:

We thank for editor’s letter and reviewers’ comments concerning our manuscript. Those comments are all valuable and certainly improve the quality of manuscript.

  1. Amplitude (y-axis) should not be adjusted 100% for each, instead shown in same AU scale (e.g. Figure 2a, but should be applied to all trace panels). Changes in amplitude are also important information and omitting such information by adjusting Y-axis scale should be avoided. GLS

Response: We updated Fig 2a, Fig 3, Fig 5d and e; Fig 6 d and e according to the comment.

  1. Overall concept is whether MM is superior to FM in physiological properties responding to drugs; in that regard, statistical significance should be examined between them, rather than comparing to basal condition of each. I would like to have all the representative traces of both AP and contractility, even they don’t have significant changes. :

Response: The statistical analysis between FM and MM is now provided in the text (Results) and graphical examples of the comparison provided in Suppl Figures.

  1. There are no N reported. How many biological/technical replicates were used in each study?:

Response: n=8 per dose/testing compound/MM or FM; n=36 for positive controls or negative controls for FM or MM. Now we also added in the text. Now added in line112-113.

 Another question is reproducibility of the experiments. In this study, the authors recorded 10-15s, applied a drug for 30 min, then recoded again. Can the authors find and record the same cells/area? If so, can the authors obtain reproducible data, especially with FluoVolt that may be bleached after imaging?

Response: We have no evidence of bleaching of FluoVolt over the duration of the experiment, in fact we routinely record Fluovolt signals repeated over 4-5 days. The dye is very stable in the membrane and the photometry is adjusted to minimize excitation light and maximize detection. We have added sentences in the Methods section was now added to reflect this point (line 128 onwards).

  1. Although authors declared no COI, they use CellOPTIQ platform from Clyde Biosciences, with which 4 of them are affiliated. This is obvious COI. Regarding the affiliations, “Janssen” is missing from #1

Response: Right. We updated and added now in the first page. 

Minor: 

  1. Line 95-96, (defing as FM) should be (defining as MM). : corrected now
  2. In Figure 1, Up90 was define, but not CtTrise that is used in other figures. Are these same?

Response:  Indeed it is not same. We take off Up90 in the figure 1.

  1. While the authors opted not to include morphological features of hiPSC-CMs on CELLvo or fibronectin, it would be helpful to add pictures/immunostaining in the manuscript, even though morphological properties have been reported. It is difficult to judge the morphology in the supplemental video.

Response: The reviewer is correct, providing repeat evidence of alignment of the cells shown in the original publication would have provided reassurance, but we were not equipped to rigorous assess myocyte structure. Our microscope did not have phase objectives, so high definition images of what would have to be mechanically quiescent images could not be provided. Visually we can observe myocytes in contracting monolayers were patterned differently from control and have included a typical video as a supplement. Our assessment was based on visual inspection to check the cell density and overall impression of the underlying cell structure. We followed the CellVo manufacturer’s instructions exactly and were using iPSC-CMs that have been previously tested by them on this system. We have modified section of the results (section 3.1) to this effect. We hope that this is sufficient for the reviewer.

  1. RNA-seq data is better to be deposited so that other researcher can explore the dataset. Response: we will provide the link of the RNA-seq data if a reviewer would like to see dataset indeed.

Response:  

Round 2

Reviewer 2 Report

The authors adequately revised manuscript.